# A globally relevant stock of soil nitrogen in the Yedoma permafrost domain

Jens Strauss [1] ✉, Christina Biasi [2], Tina Sanders [3], Benjamin W. Abbott [4], Thomas Schneider von Deimling [1,5], Carolina Voigt [2,6], Matthias Winkel [7], Maija E. Marushchak [2,8], Dan Kou [2], Matthias Fuchs [1], Marcus A. Horn [9], Loeka L. Jongejans[1,10], Susanne Liebner [11,12], Jan Nitzbon [1], Lutz Schirrmeister [1], Katey Walter Anthony [13], Yuanhe Yang [14], Sebastian Zubrzycki [15], Sebastian Laboor [1], Claire Treat [1] & Guido Grosse [1,10]

Nitrogen regulates multiple aspects of the permafrost climate feedback, including plant growth, organic matter decomposition, and the production of the potent greenhouse gas nitrous oxide. Despite its importance, current estimates of permafrost nitrogen are highly uncertain. Here, we compiled a dataset of >2000 samples to quantify nitrogen stocks in the Yedoma domain, a region with organic-rich permafrost that contains ~25% of all permafrost carbon. We estimate that the Yedoma domain contains 41.2 gigatons of nitrogen down to ~20 metre for the deepest unit, which increases the previous estimate for the entire permafrost zone by ~46%. Approximately 90% of this nitrogen (37 gigatons) is stored in permafrost and therefore currently immobile and frozen. Here, we show that of this amount, ¾ is stored >3 metre depth, but if partially mobilised by thaw, this large nitrogen pool could have continental-scale consequences for soil and aquatic biogeochemistry and global-scale consequences for the permafrost feedback.

Soils and sediments of the northern permafrost zone store a globally relevant reservoir of organic matter (OM). The amount of soil organic carbon (OC) is estimated at ~1500 gigatons (Gt, billions of tons) of OC (down to ~50 m for some deep deposits) in terrestrial environments of the permafrost region, with an additional ~400 Gt in regions with thick sediment overburden and Arctic deltas[1–4]. Global climate change is warming northern permafrost regions substantially (nearly four times) faster than the global average[5–7], and current Siberian heating is unprecedented during the past seven millennia[8], triggering widespread permafrost degradation and collapse[9,10]. Permafrost thaw is exposing currently frozen, long dormant OM to microbial processes (up to mean loss of 341 Gt (~20%) of the now frozen OM mineralised until 2300 under RCP8.5[11]) and physical mobilisation (not fully quantified yet[12,13]). Recent studies have improved our understanding of present and future permafrost OC dynamics[3,11,14,15] and highlighted the importance of abrupt and deep permafrost thaw[10,16,17]. Regions with a

[1]Permafrost Research Section, Alfred Wegener Institute Helmholtz Centre for Polar and Marine Research, Potsdam, Germany. [2]Department of Environmental and Biological Sciences, University of Eastern Finland, Kuopio, Finland. [3]Institute for carbon cycles, Helmholtz-Zentrum Hereon, Geesthacht, Germany. [4]Department of Plant and Wildlife Sciences, Brigham Young University, Provo, USA. [5]Geography Department, Humboldt University of Berlin, Berlin, Germany. [6]Département de Géographie, Université de Montréal, Montréal, Canada. [7]GFZ German Research Centre for Geosciences, Interface Geochemistry Section, Helmholtz Centre Potsdam, Potsdam, Germany. [8]Department of Biological and Environmental Science, University of Jyväskylä, Jyväskylä, Finland. [9]Institute of Microbiology, Gottfried Wilhelm Leibniz Universität Hannover, Hannover, Germany. [10]Institute of Geoscience, University of Potsdam, Potsdam, Germany. [11]GFZ German Research Centre for Geosciences, Geomicrobiology Section, Potsdam, Germany. [12]Institute of Biochemistry and Biology, University of Potsdam, Potsdam, Germany. [13]Water and Environmental Research Center, University of Alaska Fairbanks, Fairbanks, USA. [14]State Key Laboratory of Vegetation and Environmental Change, Institute of Botany, Chinese Academy of Sciences, Beijing, China. [15]Center for Earth System Research and Sustainability (CEN), Universität Hamburg, Hamburg, Germany. ✉e-mail: jens.strauss@awi.de

high amount of excess ice are especially prone to abrupt thaw and associated mobilisation of OM. A prime candidate for rapid thaw processes is the focus region of this study: the Yedoma domain in Siberia and Alaska (Fig. 1). This region consists of tens of metres of ice-rich silty soil intersected by ice wedges that developed in tundra-steppe environments of the late Pleistocene (ca. 100–12 thousand years (ka) ago). Additional deposits such as thermokarst lake and Alas sediments or Holocene cover layers started developing due to permafrost degradation and aggradation during the Late Glacial and the Holocene warm periods (ca. 14.5–0 ka ago). Because of the region's high OM content and substantial sedimentary volume, the Yedoma domain contains 327–466 Gt OC calculated down to ~20 m, representing ~26% of total permafrost OM while just covering ~12% of the northern permafrost region[4,18].

Compared to OC stocks and vulnerability, little is known about the stocks and fate of nitrogen (N) of the permafrost zone. Even though several studies have suggested globally relevant quantities of N are stored by the permafrost zone, large uncertainties remain about quantity, distribution, and vulnerability of these N stocks to climate change[19–23].

First-order estimates of permafrost soil N range from 22 to 106 Gt N with a mean of 66 Gt N[19,24]. However, these estimates only include the uppermost 3 m, excluding deeper soils and sediments such as deltaic and Yedoma deposits, which represent a large portion of OM pools that could result in N release in the coming decades[18]. It is estimated that abrupt thaw processes could affect 1.6 million km² by 2100, impacting half of permafrost OM through collapsing ground, rapid erosion, and landslides[10]. Especially Yedoma landscapes are one of the major reasons why so much permafrost carbon is stored in areas classified as "very high" risk of abrupt thaw and surface collapse in the future[25]. Because ice wedges extend through the full depth of Yedoma

deposits (Fig. 2), permafrost thaw triggers surface collapse as well as fluvial and coastal erosion[10,26,27]. The melting of ice can also create macropores and microtopography that alter soil hydrology, potentially mobilising soil N from well below the surface[27]. Thus, also deep mobilisation of OM is expected to intensify on the short- to mid-term term (coming decades[28] to end this century[10]), in total up to twelve-fold faster than expected[29]. However, exact numbers are missing as in most models still the abrupt thaw processes are not implemented.

Nitrogen availability regulates key components of the OC cycle at high latitudes[30–32] and the mineral N cycle provides the substrate for microbial production of nitrogenous gases, most importantly nitrous oxide ($N_2O$), which is a potent, ozone-depleting greenhouse gas with a global warming potential approximately 300 times greater than $CO_2$ on a centennial timescale[33]. $N_2O$ production and release represents an only recently recognised non-carbon climate feedback from thawing permafrost[34,35]. Consequently, reducing uncertainty about permafrost N stocks is critical for improving estimates of the overall magnitude and timing of the permafrost climate feedback[11,36]. Permafrost N stocks have the potential to influence the permafrost climate feedback in four interrelated ways. First, increased N availability in terrestrial ecosystems following permafrost degradation can stimulate primary productivity[37,38]. Generally, permafrost-affected soils are assumed to have a strong mineral N-limitation for microbes and plants due to low mineralisation rates in cold soils[39,40]. However, a recent synthesis showed that gross ammonification and nitrification rates in active layers were of similar magnitude as observed in temperate and tropical systems[41]. Instead, the rather short period when soils are not frozen seems to be the main factor limiting N turnover. Nevertheless, increased availability of N may lead to $CO_2$ sequestration through plants given that they successfully compete with microbes for available nutrients[42]. Second, increased N availability can affect OM

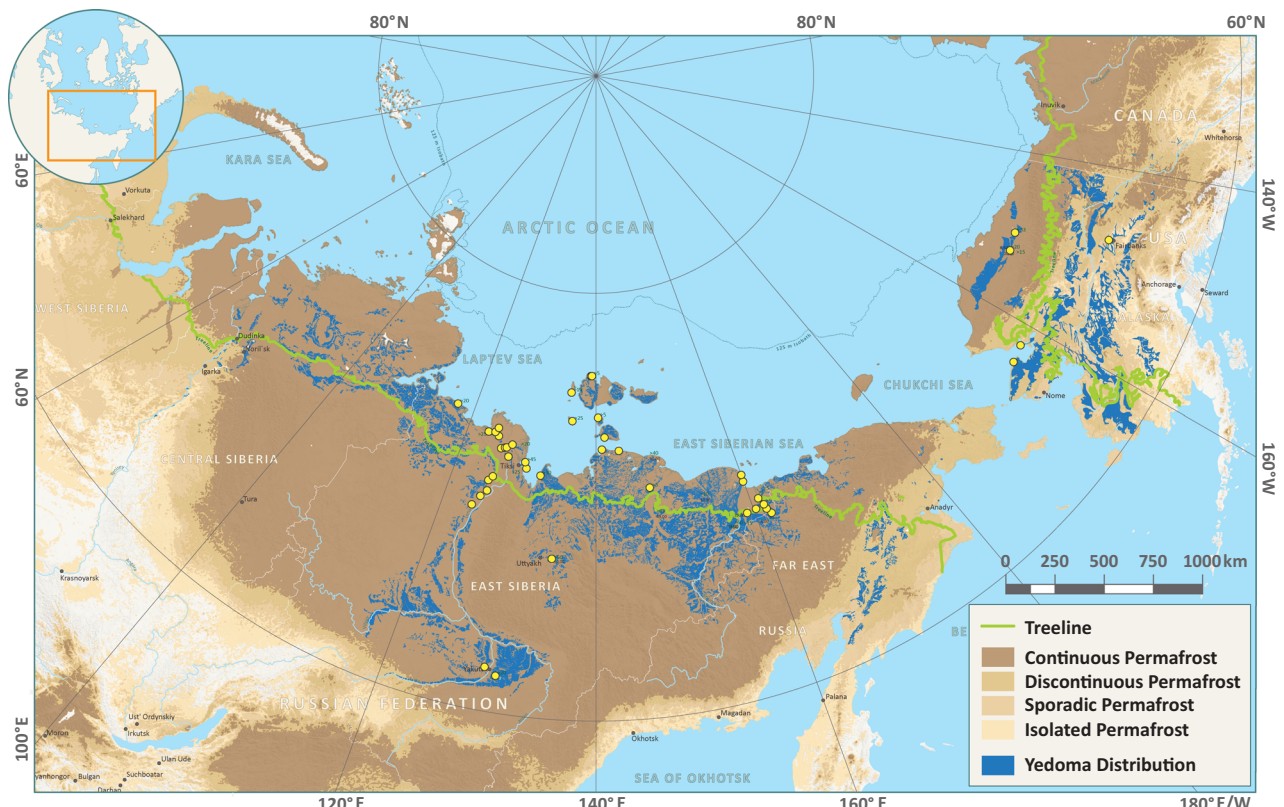

**Fig. 1 | Spatial distribution of the Yedoma domain in the northern high latitudes.** Blue areas illustrate the potential occurrence of Yedoma deposits, also including the other mentioned types of deposits (active layer, Holocene cover, thermokarst and Taberite deposits). The yellow dots show the study sites. Map adapted from Strauss et al.[59] with datasets from Obu et al.[84] for the permafrost coverage and Walker et al.[85] for the treeline.

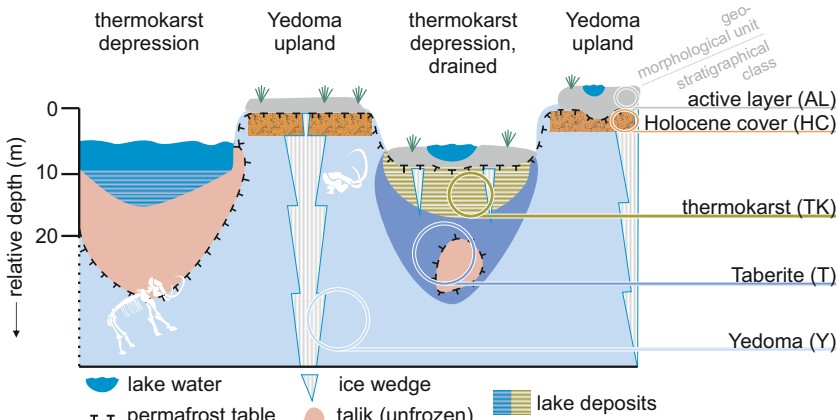

**Fig. 2 | Conceptual model of the stratigraphic position and degradation legacy.** Conceptual model of the stratigraphic position and degradation legacy (former freeze-thawing history) of the considered deposits: active layer (grey colour code, $n = 467$), Holocene cover (orange, $n = 175$), thermokarst deposits, (olive-green, $n = 479$), Taberite (in-situ thawed, diagenetically altered but not relocated former Yedoma deposits, bluish-purple $n = 175$), and Yedoma (ice-blue, $n = 917$) deposits. The colour code is used in Figs. 3, 4. The depth is approximate and meant to communicate the general depth of different landscape units. For the depth-specific nitrogen estimates, absolute depth measurements are used. Abbreviations introduced here are used in all other figures and tables: AL active layer, HC Holocene cover on top of Yedoma, TK Thermokarst deposits, T Taberites, Y, Yedoma deposits.

decomposition. While some studies found that greater N availability can accelerate OM decomposition[30,32], at least for the first years after thaw[43], others show the opposite effect thereby either amplifying or reducing the climate feedback depending on the net interaction[11,44–47]. Third, increased N availability in combination with changes in soil redox conditions and vegetation disturbance can stimulate microbial production of $N_2O$ during and after permafrost degradation[21,36,48–52]. High $N_2O$ emissions were recently found at Yedoma thawing sites[35] putting N reservoirs in the Yedoma domain high on the agenda in Arctic soil and climate change research. In addition to the gaseous N loss as $N_2O$, release of the final denitrification product dinitrogen ($N_2$) could further enhance N limitation in northern soils. Fourth, permafrost degradation increases lateral export of organic and inorganic N to aquatic ecosystems[53–55]. As for terrestrial ecosystems, increased nutrient availability can alter aquatic and marine food webs, affecting ecosystem C balance and biodiversity[21,56–58]. To understand the magnitude of these processes, it is important to assess the quantity and vulnerability of permafrost N stocks to climate change.

In study our specific goals are to estimate the N pools across different stratigraphic units of Yedoma domain permafrost soils and deposits (Fig. 1) to enhance the understanding of potential climate feedbacks. Therefore, we analyse 2,213 soil samples from the Yedoma domain in Alaska and Siberia to assess N distribution and content. Additionally, based on a subset of 1200 samples, we synthesise the availability of mineral N forms (ammonium ($NH_4^+$) and nitrate ($NO_3^-$)), to assess potential impact of N released from decomposing OM. Here we show that the Yedoma domain contains a globally relevant N stock of $41.2^{+3.3}_{-3.9}$ Gt N with 90% of which occurs in currently inaccessible permafrost. These large N stock also appear to be highly bioavailable based on their low C:N ratios and relatively high proportion of inorganic forms of N. As climate change degrades Yedoma landscapes, a portion of this N will be activated and will affect the permafrost-climate feedback.

## Results
### Quantity of N in deposits of the Yedoma domain
To reach the highest possible sample coverage and representativeness, our results are based on 1,092 samples from late Pleistocene deposits and 1121 samples from Late Glacial and Holocene deposits (Fig. 2).

We found that the Yedoma domain deposits store a globally significant pool of N that may partially become activated following permafrost thaw. In total, the Yedoma domain contains $41.2^{+3.3}_{-3.9}$ Gt N (median with 5th and 95th percentiles) in the studied five stratigraphic units (Table 1). 90% of these N stocks ($37.0^{+3.1}_{-3.8}$ Gt N) are stored in perennially frozen soil (i.e., permafrost). Three major sub-reservoirs compose this N pool: thermokarst ($15.4^{+2.0}_{-2.1}$ Gt N), Taberite ($13.8^{+2.1}_{-2.9}$ Gt N) and Yedoma deposits ($6.6^{+1.1}_{-1.2}$ Gt N). The active layer in the Yedoma domain stores $4.2^{+1.0}_{-1.1}$ Gt N, and the Holocene cover on top of Yedoma deposits (excluding active layer) stores $1.2^{+0.2}_{-0.2}$ Gt N. Of the total N stock of the Yedoma domain, approximately a quarter is contained in the top 3 m, with the majority stored below that depth (Table 2).

### Classification of the of N stocks
Classification of the N pool according to soil stratigraphic age revealed that the late Pleistocene deposits contain $20.4^{+2.4}_{-3.1}$ Gt N (-49.5 %, Table 2), and that Holocene soils and sediments contain $20.8^{+2.2}_{-2.4}$ Gt N (-50.5 %). Classified into the categories 'seasonally thawed' (active layer soils, Table 2), 'thawed in the past' (Holocene cover, thermokarst deposits, Taberites), and 'never thawed' since deposition (Yedoma) results in a 10: 74: 16 ratio.

### N, $NH_4^+$ and $NO_3^-$ density and the C:N ratio
The largest N density occurred in the thermokarst basin deposits ($2.2^{+0.3}_{-0.3}$ kg N m$^{-3}$), followed by the Holocene cover ($1.9^{+0.3}_{-0.3}$ kg N m$^{-3}$), active layer ($1.6^{+0.4}_{-0.4}$ kg N m$^{-3}$), Taberite ($1.5^{+0.2}_{-0.3}$ kg N m$^{-3}$) and Yedoma deposits ($0.9^{+0.1}_{-0.2}$ 0.9 kg N m$^{-3}$) (Fig. 3). This pattern suggests that the N density is substantially altered from Yedoma to Taberite sediments (i.e. after thawing). This relative enrichment is explained by the Taberite genesis: the melting of excess ice increases the bulk density (BD) of the sediment, resulting in higher N densities.

The total OC to N (C:N) ratios were intermediate to low, decreasing from a median of 18.8 in Holocene deposits to ~9.2 in Taberites and Yedoma deposits (Table 1, Fig. 4). The opposite was true for mineral N pools, determined from pore water concentrations of $NH_4^+$ and $NO_3^-$. The highest densities were found in Taberites (for $NO_3^-$, no data available for $NH_4^+$) and Yedoma deposits (for both $NH_4^+$ and $NO_3^-$), and the lowest in the active layer (Fig. 4). Importantly, the high mineral N density in Yedoma was associated with a low total N density compared to the different stratigraphic units, indicating a high proportion of reactive N species in total soil N.

**Table 1 | Parameters for calculating the N budget**

| Stratigraphical class | Mean thickness (m) | Coverage (km²)ᵃ | Mean bulk density (10³ kg m⁻³) | Wedge-ice content (vol%) | Mean TN (wt%) | C:N | N stock (Gt) | + | – | % | N density (kg m⁻³) | + | – | Thawing history | Depositional age |
|---|---|---|---|---|---|---|---|---|---|---|---|---|---|---|---|
| AL | 1ᵃ | 2,586,825 | 0.63 | 0ᵃ | 0.53 | 18.8 | 4.2 | 1.0 | 1.1 | 10.2 | 1.6 | 0.4 | 0.4 | seasonally thawed | Holocene |
| HC | 2 | 319,827 | 0.72 | 7 | 0.48 | 15.2 | 1.2 | 0.2 | 0.2 | 2.9 | 1.9 | 0.3 | 0.3 | thawed in the past | Holocene |
| TK | 6 | 1,460,378 | 0.73 | 7 | 0.56 | 11.5 | 15.4 | 2.0 | 2.1 | 37.4 | 2.2 | 0.3 | 0.3 | thawed in the past | Holocene (and Late Glacial) |
| T | 6.1ᵃ | 1,460,378 | 0.94 | 0ᵃ | 0.20 | 9.2 | 13.8 | 2.1 | 2.9 | 33.5 | 1.5 | 0.2 | 0.3 | thawed in the past | late Pleistocene |
| Y | 20 | 479,741 | 0.87 | 45 | 0.36 | 9.3 | 6.6 | 1.1 | 1.2 | 16.0 | 0.9 | 0.1 | 0.2 | 'never' thawed | late Pleistocene |
| Total | | | | | | | 41.2 | 3.3 | 3.9 | | | | | | |

The budget uncertainties are the 5 and 95 interquartile range around the mean. See Fig. 1 for the abbreviations of the stratigraphical classes.
ᵃUsed as a fixed parameter that is not resampled during the bootstrapping procedure.

## Discussion

Yedoma domain permafrost soils and deposits store much more N than previously recognised. While the Yedoma domain only comprises ~12% of the global permafrost zone[59], our estimate (41.2 Gt N) of permafrost N in Yedoma is larger than what has been previously suggested as the lower range of N stock estimates for the entire permafrost region (31 to 102 Gt N, mean 66 Gt[19]). Furthermore, the soil N stock in the Yedoma domain is more than four times as large as the soil N stock in northern peatlands (>23° latitude; 10 Gt N[60]), although the soil OC stocks are similar (327 to 466 Gt OC in Yedoma deposits, 415 Gt OC in northern peatlands). This is a consequence of the C:N ratios in the deposits of the Yedoma domain (9 to 19) compared to peatlands (24 to 35)[60], highlighting the importance of Yedoma deposits for the total N stock of northern soils.

While there is some overlap between our first 3 m N stock estimate for the Yedoma domain and the earlier N stock estimates for the global permafrost zone (66 Gt[19]), just the deep deposits alone (Taberites and Yedoma deposits) contain $30.6^{+3.1}_{-3.8}$ Gt N. This increases the current estimates of total permafrost N stocks by 46% to 97 Gt N (66 Gt[19] + ~31 Gt (>3 m, this study). These stocks represent a substantial revision to estimates of global soil N, which were previously around 150 Gt N total, including global permafrost[61]. In addition to updating estimates of N quantity, this study provides valuable information about the type and bioavailability of these N stocks as reflected by C:N ratios and mineral N content. Despite the relatively low N density compared to the other stratigraphic units, permafrost deposits in the Yedoma domain held a substantial amount of reactive, inorganic N in the forms of $NH_4^+$ and $NO_3^-$ (Fig. 4). High mineral N content has also been observed in (and outside[62]) Yedoma domain permafrost previously (of $NH_4^+$ in particular, but also of $NO_3^-$)[43,44,51,56], which could be due to a combination of OM decomposition prior to permafrost aggradation, during the frozen period, or after permafrost degradation. For example, higher inorganic N concentrations have been observed at the permafrost-active layer interface[63], as well as during fall and spring microbial turnover periods[55,64,65]. Additional factors contributing to the low C:N ratios in the Yedoma and Taberite sediments could be related to the pedogenesis: formation of these soils included the rapid burial and freezing of plant remains composed mostly of grassland plants with low C:N ratios.

Looking at the influence of thaw processes in the Yedoma domain on N stocks we know that thaw processes in the Yedoma domain can activate deep soil deposits through mechanisms such as thermo-erosion, coastal collapse, fluvial erosion, and thermokarst lake formation[12,16,66]. Besides being confident that models considering only gradual permafrost thaw are substantially underestimating carbon emissions from thawing permafrost[10,16,29], no quantitative circum-Arctic estimate on future thaw depth and coverage are available. The high ground ice content, (~80 vol%[18,67,68]) and the large ice wedges extending through the full depth of deposits make Yedoma particularly susceptible to abrupt thaw. For example, sediments beneath newly formed thermokarst lakes showed permafrost degradation to depths of 10–15 m within 50–60 years[17], increasing the total volume of thawing soil beyond what is visible in changes in lake area. Thawing of permafrost persists for some time following thaw lake drainage[69], which also indicates a disturbance to permafrost that can be more widespread across the landscape than observed by the expansion of thaw lakes. In this study, this long-lasting legacy of permafrost thaw is evident deep in the soil profile from observed differences between Yedoma and Taberite sediments that increased the N density between the soils of the same origin (Fig. 3).

While accurate pan-Arctic estimates of the current abundance of these abrupt and deep thaw features are lacking due to difficulties in detecting permafrost degradation from large-scale remote-sensing data, estimates of total lake area change driven by permafrost thaw in recent decades in ice-rich permafrost areas range from 3% to 8%

between 1999 and 2015[70] With ongoing heat waves and wildfires (boreal forest and tundra fires) causing an initial disturbance more deep thermokarst processes[71,72] are very likely. Similarly, talik formation and unfrozen soil not located under thaw lakes cannot be detected using remote-sensing methods, though this mode of permafrost degradation appears to be widespread[27]. Further, widespread surface subsidence by the loss of excess ice[73], thermo-erosion and other thaw slumping associated with abrupt thaw dramatically changes the landscape, making formerly deep deposits (>3 m) accessible even for shallow rooting plants, which can potentially utilise fractions of large N stocks in deep deposits[37,38,51]. A first-order estimate following the approach of Nitzbon et al.[29] suggests that by 2100 an additional 0.2–0.8 Gt N could be affected by thaw in the Yedoma region under an ambitious mitigation scenario (RCP2.6), and 4.3–16.3 Gt N—about 40% of the total N stock estimate—under a high emission scenario (RCP8.5) (Fig. 5). In comparison, the weathering of near-surface rocks globally are estimated to mobilise 0.02 to 0.003 Gt N, annually[74], indicating that newly thawed Yedoma N could potentially be a large, non-anthropogenic additional input to the global N cycle.

If available to plants, this newly thawed N of up to ~16 Gt (high emission scenario) could have a fertilisation effect and increase plant growth. However, in a modelling study[75] this fertilisation effect was recently found to be much weaker than expected in permafrsot ecosystems. This was likely caused by a significant mismatch between the timing of peak plant growth (early to mid-summer) and peak thaw depth (late summer to fall) that resulted in incomplete plant use of N deeper in the profile, near the permafrost table. The increased N availability enhanced the N loss pathways, leading to increased $N_2O$ emissions in the applied model[75]. With our data, this could mean that the deeper the thaw, the greater the temporal mismatch and increasing potential for $N_2O$ release. In field conditions, Marushchak et al.[35] recently observed large $N_2O$ release from eroding Yedoma domains a few years post-thaw. The $N_2O$ release occurred after slope stabilisation following abrupt permafrost thaw, potentially because of the combination of drying, the release of inorganic N from decomposing OM, and the establishment of fungal and plant communities. These favourable conditions for overall gaseous N losses and $N_2O$ production are most likely to prevail in Taberite and Yedoma deposits, which comprise nearly half of the total N stock of the Yedoma domain (49%). As mentioned before, not all Taberite and Yedoma deposits all across the full depth range will be exposed if thawed. Nevertheless, they have low C:N ratios, high mineral N pools (Fig. 4) and labile OM in it, all of which are key factors that can result in $N_2O$ production and emissions[34,35].

There is some evidence of N mobility and loss from Yedoma soils to aquatic ecosystems with permafrost thaw. Walter Anthony et al.[31] found strongly elevated dissolved inorganic N concentrations in Yedoma thermokarst lakes compared to Arctic lakes outside the Yedoma domain or to floodplain lakes in the Yedoma domain. Widespread and rapid thermokarst and thermo-erosion across the Yedoma domain also leads to massive hydrological transport of OM to rivers, lakes, estuaries and Arctic shelves seas[76–78]. In their study of the Lena

**Table 2 | Classification of the N pool according to age, depth and thaw legacy**

|  | N budget (Gt) | + | − | % |
|---|---|---|---|---|
| Frozen | 37.0 | 3.1 | 3.8 | 89.8 |
| Unfrozen | 4.2 | 1.0 | 1.1 | 10.2 |
| Holocene | 20.8 | 2.2 | 2.4 | 50.5 |
| Pleistocene | 20.4 | 2.4 | 3.1 | 49.5 |
| Seasonally thawed | 4.2 | 1.0 | 1.1 | 10.2 |
| Thawed in the past | 30.4 | 2.9 | 3.6 | 73.8 |
| 'Never' thawed since deposition | 6.6 | 1.1 | 1.2 | 16.0 |
| Below 1 m | 36.4 | 3.1 | 3.8 | 88.3 |
| Below 3 m | 30.6 | 3.1 | 3.8 | 74.2 |

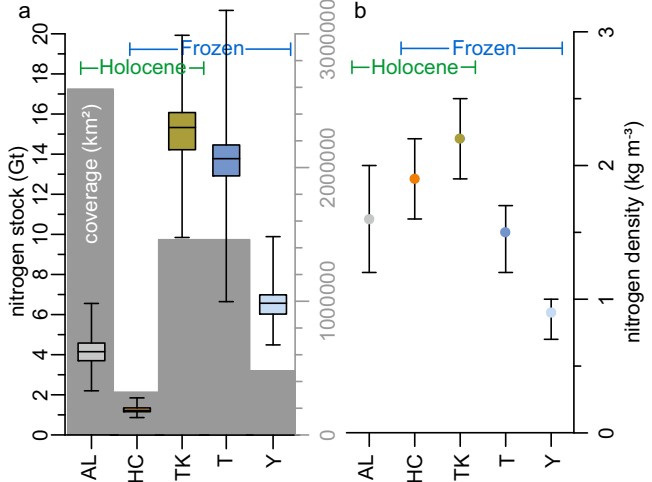

**Fig. 3 | Assessment of N stocks of the Yedoma domain, subdivided into five stratigraphic units. a** N stock boxplots with min-max ranges and areal coverage of each unit (grey vertical bars) in km². **b** N density in kg N m⁻³. See Fig. 1 for the abbreviations of the stratigraphical classes.

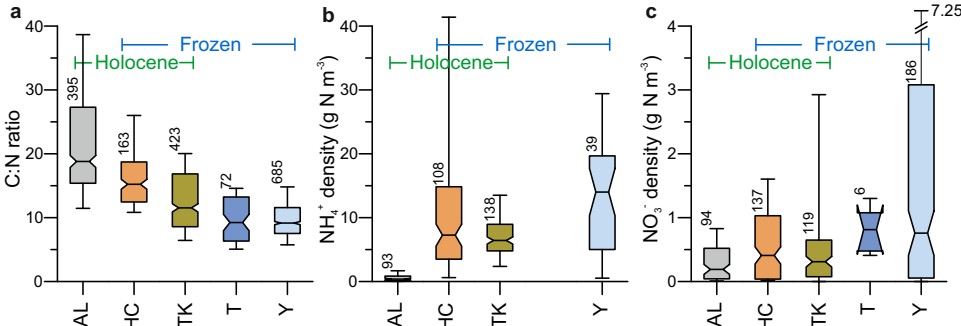

**Fig. 4 | N biovailability indicators.** The C:N ratio (**a**), $NH_4^+$ (**b**) and $NO_3^-$ (**c**) density of the different stratigraphic units. Whiskers show the 10/90 quantiles, the boxes are bordered by the 1st and 3rd quartiles, including the median with notches (95% confidence interval around the median). The numbers above the whiskers show the number of samples. For calculating C:N ratios, total N values <0.05 were excluded to avoid calculation artefacts due to measurement uncertainties. For visualisation of the $NO_3^-$ density of Yedoma, we use a broken scale to show the very high upper value (7.25). See Fig. 1 for the abbreviations of the stratigraphical classes.

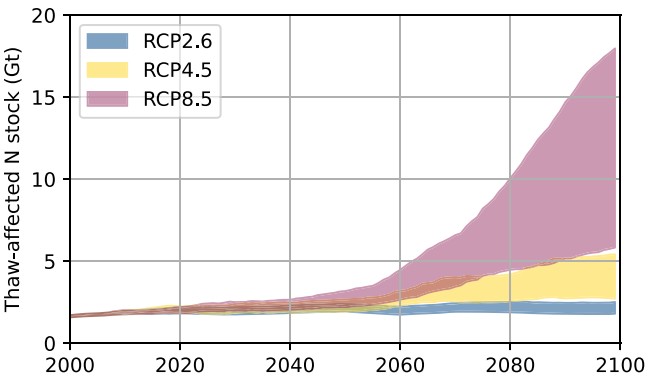

**Fig. 5 | Projection of thaw-affected N stocks.** The model estimate for thaw-affected C stocks by Nitzbon et al.[29] was adjusted to our N stocks by scaling simulated thaw depths under consideration of excess ice melt with the total areal extent of three geomorphological units composed of five stratigraphical classes (see Table 1, Fig. 2). For each warming scenario the indicated ranges correspond to 11-year running means of the annual maximum of thawed N under contrasting hydrological conditions.

River to nearshore, Fuchs et al.[12] and Sanders et al.[79] showed that there was a significant amount of reactive N released to the river system, likely also from Yedoma deposits in the catchments.

The thawing of these N-rich sediments can link to C cycling by enhancing both primary productivity in terrestrial and aquatic ecosystems[37,79]. Increased N availability also affects soil decomposition processes, potentially offsetting enhanced vegetation gains[32]. Therefore, changes in N availability can potentially affect the C balance in the Yedoma domain through these feedbacks but the net effect is unclear.

In conclusion, we found the Yedoma domain contains a globally relevant N stock. This N has accumulated in the Yedoma domain due to rapid burial and freezing of plant remains composed mostly of plants with low C:N ratios. In total, the Yedoma domain contains $41.2^{+3.3}_{-3.9}$ Gt N in the studied five stratigraphic units, 90% of which (37.0 Gt) occurs in perennially frozen sediments, where it is currently inaccessible. Assuming a high emission scenario, 4–16 Gt of the N (up to 40%) could become available by thaw until 2100. These large N stocks also appear to be highly bioavailable as we found that the quality of Yedoma domains' OM is making it vulnerable/favourable for mobilisation, as its N is potentially highly bioavailable, as indicated our results of high N content, low C:N ratio and low DON:mineral N ratio. As climate change degrades Yedoma permafrost deposits, a portion of this N could be activated, increasing local greenhouse gas release and affecting regional C balance in terrestrial, aquatic, and marine ecosystems. This could lead to additional $N_2O$ emissions, but also it could mitigate climate feedbacks through promoting enhanced vegetation C sequestration. In all cases, there is strong evidence that the permafrost-climate feedback will be affected by the amount and state of mobilisation of this previously unquantified N pool.

## Methods
### Material
We included 1092 samples from late Pleistocene deposits and 1121 samples from Late Glacial to Holocene deposits. The thermokarst processes started during the Late Glacial warming (ca, 14,670 to ca. 12,890 years BP) and intensified during the Holocene thereafter. As the majority of the samples from thermokarst is from the Holocene, we referred to this sample category as Holocene. The Holocene deposits included 467 samples from the active layer (seasonally thawed layer) above permafrost soils, 175 samples from perennially frozen Holocene cover deposits accumulated on the top of Yedoma, and 479 samples from refrozen thermokarst deposits in drained thermokarst lake basins (Fig. 2). The late Pleistocene deposits included 175 samples from

in situ thawed, diagenetically (anaerobic microbial decomposition possible during unfrozen phase) altered Yedoma deposits, which are refrozen or still thawed (called Taberite), and 917 samples from frozen Yedoma deposits. The depth with a complete outcropped sediment unit (full range, top to bottom) was taken from 21 sites for Yedoma deposits (range 5 – 41 m, mean 20 m, median 19 m) and 9 sites for thermokarst deposits (range 2 – 13 m, mean 6 m, median 5 m). For most Yedoma sites, the base was not reached, so we were just able to give a minimum value.

### Coverage of Yedoma domain
We defined our study area as the maximum, historic coverage of Yedoma deposits (Fig. 1) excluding marine inundated shelf regions[59].

### Laboratory measurements and calculations
In total, 2213 individual sediment samples were collected in the time span of 1998–2018 from permafrost soils from 42 separate locations, natural exposures along coasts and rivers, shallow boreholes, and deep (>20 m) permafrost cores.

TN content of 1784 samples and the TOC (carbonate removed) content of 2213 samples were measured using an elemental analyser (Vario El III, CS-Autoanalyser ELTRA CS 100/1000S; CNS Micro-analyser, LECO 932) after being freeze dried. The C:N ratio was calculated from the TOC and TN measurements. All measured TN values are given in weight percentage wt%.

BD were measured directly with a standard cylinder or, for the permafrost and taberite sediments, calculated using the ice content by assuming ice saturation following Strauss et al.[80].

### $NH_4^+$ and $NO_3^-$ quantification
Moreover, we established a database associated with mineral N pools (here: $NH_4^+$ and $NO_3^-$) for different stratigraphic units across the Yedoma domain. For this, we compiled published and non-published data. In total, the data consists of 658 samples, including 378 data points for $NH_4^+$ (active layer, 93; Holocene cover, 108; thermokarst sediment, 138; Taberite, 0; Yedoma deposit, 39) and 542 data points for $NO_3^-$ (active layer, 94; Holocene cover, 137; thermokarst sediment, 119; Taberite, 6; Yedoma deposit, 186). The data represent mostly mineral N concentrations in the pore water from thawed sediments reported as mg N L$^{-1}$, obtained by rhizon pore water samplers or pore water presses. There are also some data points representing $NH_4^+$/$NO_3^-$ concentrations in ice wedges, which were measured by ion chromatography. Since the concentrations in the pore water are dependent on soil water content, which varies greatly between stratigraphic units, we converted the data into $NH_4^+$/$NO_3^-$ density values (g N m$^{-3}$ soil) to better relate them to the total soil N content. For the conversion, we used $NH_4^+$/$NO_3^-$ concentrations (mg N L$^{-1}$), mean BD of each stratigraphic unit (Table 1), and mean soil water content of each stratigraphic unit (grav-%, calculated based on the mineral N database). In detail, we used following mean values derived from data published by Schirrmeister et al.[81] for each class (given as gravimetric (related to dry basis, grav-%) and absolute (relates to wet weight, abs-%): AL: 56 grav-%| 36 abs-%; H: 84 grav-%| 46 abs-%; TK: 209 grav-%| 68 abs-%; T:31 grav-%| 24 abs-%; Y: 77 grav-%| 44 abs-%. With this dataset, we compared the mineral N pools among stratigraphic units, and analysed the relationship between $NH_4^+$/$NO_3^-$ and C:N ratio.

### Statistical methods
We used statistical bootstrapping techniques to resample observed values for a more robust estimate of N stocks[82]. A total N pool estimate was derived for each of the 10,000 bootstrapping runs, providing an overall estimate of mean and variance. The number of resampling steps for each parameter is connected to the original number of observations of the different parameters deposit thickness, N content, BD, and wedge ice. Because N content and BD of individual sediment

samples are correlated, paired values were used in the resampling process. We used this relationship to calculate the BD using an exponential function BD = $1.0341 \times e^{(-0.062 \times TOC)}$ ($r^2 = 0.67$) that was fitted to those data for which both values were available. We used the $R$ statistical computing program for all computations.

Generally, we calculated the total N stock as follows:

$$m_{TN,tot} = \sum_{i=1}^{n} m_{TN,i} = \sum_{i=1}^{n} d_i(1 - f_{wedge,i})\rho_{b,i}c_{TN,i}A_i \qquad (1)$$

$m_{TN,tot}$: pool of total $N$ in the included units,

$i = 1...n$: index numbers of the single unit considered for the $N$ budgeting,

$d_i$ deposit thickness of unit $i$,

$f_{wedge,i}$: volume fraction of ice wedges in unit $i$,

$\rho_{b,i}$: bulk density of deposits of unit i, paired with $c_{TN,i}$,

$c_{TN,i}$: content of total nitrogen (TN) in deposits of unit i, paired with $\rho_{b,i}$,

$A_i$: area of unit $i$.

The error estimates in this study represent the 5th and 95th percentiles. We estimate the N stocks for each unit separately, and then sum them up for the total N stock estimate.

## Projection of thaw-affected nitrogen stocks

We followed Nitzbon et al.[29] to estimate the portion of the N stocks that could become subject to thawed conditions within the course of the current century. For this, we took the maximum thaw depths simulated by Nitzbon et al.[29] for two different geomorphological units (Thermokarst lake basins and Yedoma uplands) using climatic forcing data for the central Lena River delta under three different emission scenarios (Representative Concentration Pathway (RCP) 2.6, RCP 4.5, and RCP 8.5) and under contrasting hydrological conditions (water-logged and well-drained). The simulations took into account rapid thaw processes due to the presence of excess and wedge ice. We multiplied the simulated thaw depths with the areal coverage of the respective geomorphological units and the N densities of the corresponding stratigraphic classes that would be affected by thaw (Table 1).

## Data availability

All data generated in this study are available here: https://doi.org/10.1594/PANGAEA.948079 (PANGAEA repository)[83].

## Code availability

The bootstrapping code we adjusted for this study is available from https://doi.org/10.5281/zenodo.3734247. The code is published under a GNU General Public License v3.0.

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

## Acknowledgements
This study was a result of "The Yedoma Region" Action Group funded by the International Permafrost Association and is embedded into the Permafrost Carbon Network (PCN). We acknowledge the support by the European Research Council PETA-CARB (#338335), the German Federal Ministry of Education and Research (BMBF; CACOON (#03F0806A), EISPAC (#03F0809A), PermaRisk (#01LN1709A), Laptev Sea System 2000, Permafrost Dynamics in the Laptev Sea, CarboPerm (#03G0836A), and KoPf (#03F0764), the German Research Foundation (HE 3622/16-1, DI 2544/1-1, WE 4390/7-1, UL426/1-1, KI 849/4-1, HO4020/3-1, HO4020/5-2), and the Initiative and Networking Fund of the Helmholtz Association (#ERC-0013). The work of C.B., C.V., M.E.M. and D.K. was supported by the Academy of Finland/Russian Foundation for Basic Research project NOCA (decision no. 314630), the Yedoma-N project (General Research Grant from the Academy of Finland, decision number 287469) and the N-PERM project (General Research Grant from the Academy of Finland, decision number 341348). C.V. was further supported by the Academy of Finland project MUFFIN (decision no. 332196), and MEM by the Academy of Finland project PANDA (decision no. 317054). B.W.A. was supported by the U.S. National Science Foundation (award no. 1916565).

## Author contributions
J.S. designed the study and generated the first draft further improved by C.B., T.S., B.W.A, M.M. and C.V. M.W. and D.K. worked on the $NH_4^+$ and $NO_3^-$ data, and L.J. conducted the bootstrapping method, J.N. modelled the future N thaw. S.L compiled Fig. 1, J.S. designed Figs. 2 and 4, Fig. 4 was drawn by D.K. and Fig. 5 by J.N. Supplementary to the previously mentioned authors, T. S.v.D., M.F., M.A.H., S.L., L.S., K. W.A., Y.Y., S.Z., C.T. and G.G. contributed significantly by revising and reviewing the manuscript drafts and/or contributing data.

## Funding

## Competing interests
The authors declare no competing interests.
