## [Peer Review File · Nature Communications]

A globally relevant stock of soil nitrogen in the Yedoma permafrost domainREVIEWER COMMENTS

Reviewer #1 (Remarks to the Author):

This study provides a detailed account of the nitrogen content in Yedoma permafrost-affected soils. Yedoma occurs in areas across Siberia and Alaska and this permafrost type is characterized by being ice-rich and containing organic matter that is considered relatively unprocessed and labile. It is estimated that the Yedoma permafrost domain stores about $\frac{1}{4}$ of the total permafrost carbon stock, while its nitrogen storage has been far less studied.

In this study, the authors impressively make use of more than 2000 samples, collected and analyzed from diverse landscapes and stratigraphic classes, to provide an updated estimate of total nitrogen and dissolved mineral nitrogen forms (NH_4 and NO_3) upscaled across the Yedoma domain to soil depths of up to 40 meters. I assume this max soil depth based on Figure 2 but I did not find it explicitly stated, while mean thickness for Yedoma class permafrost is given as 20 meters. The methods used and the resulting data generally appear robust, see specified comment below.

I believe this study will be a solid contribution to several research fields focusing on permafrost-affected landscapes. The amount of nitrogen stored in the Yedoma domain is large and I agree with the authors that this nitrogen stock is relevant in a global perspective and when considering various ecosystem-climate feedback mechanisms. However, the question remains how much of this permafrost nitrogen will become biologically available and mobilized in the future. The authors often convey a message of the Yedoma permafrost nitrogen stock to be vulnerable to thaw. Nevertheless, $\frac{3}{4}$ of the estimated nitrogen stock lies frozen below 3 meters and up to a max depth of several tens of meters. Labeling near-surface and deep permafrost layers together as “vulnerable to thaw” seems somewhat unrealistic for the permafrost landscapes that are not erosion sites, e.g. not coastal or riverbed cliff areas. I believe that the manuscript will improve with some added text focusing on the extent of soil layers or depths, perhaps of specific landscape types, which are estimated to thaw over a certain period. This timeframe could be short- or long-term. Without some quantification of projected thaw trajectories and a timeframe, I find it difficult to assess the climate-feedback importance of the large nitrogen stock found at depth.

Please see additional detailed commentary below.

Specific comments:

L16: Please add info on the depth limit of this nitrogen estimate.

L19: I suggest reporting that $\frac{3}{4}$ of the Yedoma permafrost nitrogen is stored at depths below 3 m, or, conversely, report that $\frac{1}{4}$ is stored above 3 m. It would be strong to refer to estimates on which soil depth groups that will likely thaw within a certain timeframe. This could be short and/or long-term but it should be defined. Stating that permafrost nitrogen, which in this study is estimated down to several tens of meters, is vulnerable to thaw across the entire depth range seems rather i) unrealistic, and ii) vague.

L27: Please provide info on the depth limit of the carbon estimate here.

L32: Please specify what you mean by 4-6 % until 2300. It is not clear from the present text.

L44: Please provide info on the depth limit of the carbon estimate here.

L54: Could you please add some details on the areal extent of deep soil (> 3 m) vulnerability to thaw in the coming decades (i.e. short-term outlook)?

L74: These studies (ref.: 39 and 40) do not focus on permafrost-affected soils.

L85/research goal ii: I agree that assessing quantity and vulnerability of permafrost nitrogen is timely

and important. However, I am not convinced that the present study fully achieves the “vulnerability assessment” aspect. Without some quantification of the extent of both shallow and deep permafrost (> 3 m) nitrogen that is estimated to thaw by a specific time, some of the statements about substantial climate feedbacks seem speculative.

Research goal ii states that the goal of this study is to “... improve our understanding of the susceptibility of these N reservoirs to thaw and climate change”. While the study successfully quantifies Yedoma nitrogen stocks (research question i), the remaining text generally reviews literature on the lability of Yedoma organic matter and possible ecosystem effects of nitrogen mobilization. This seems a somewhat different focus than what is actually stated in research goal ii. I suggest changing “improve” to “review” in research goal ii.

Figure 1: When printed (but also to some extent digitally), it is not always easy to distinguish the green colour for “tree line” from the blue Yedoma distribution range. Consider modifying colour scheme.

Figure 2: This is a nice overview conceptual figure. I recommend inverting the Y-axis so that 0 m is at the ground surface. Please provide some explanation for the gray thermokarst colour class, and explanation for the gray dashed line that shows the extent of the Taberite class. How are thermokarst depressions used in the depth calculations, do they begin at several meters of depth or do you count the surface of thermokarsts as 0 meter, similar to Yedoma upland? This info seems relevant for the depth-specific nitrogen estimates. Note that Taberite is supposedly purple according to the text but appears blue on my screen and in print. Using the colour scheme here for the subsequent figures and tables works well. However, I encourage the authors to use a colourblind-friendly colour scheme. I do not think that the “active layer” class needs to be red if thermokarst is green.

L177: Are there any model estimates in recent literature of how much of this deep permafrost (>3 m depth) that will be exposed to thaw over a specified timeframe? That seems relevant here for the importance of the deep soil nitrogen pools.

L180: But that depends on how thaw occurs, right? With thaw collapse as in retrogressive thaw slumps then shallow and deeper permafrost layers may be mixed and some of the deep Yedoma nitrogen could contribute to relatively high N₂O fluxes as in the study by Marushchak et al. However, in other landscapes, where thaw slumping does not occur as drastically then deeper permafrost layers will likely stay buried and anoxic, which may not promote N₂O production. See also comment for L282.

L197: Please check your values, plant nitrogen per m² should not be this high. In addition, why compare Yedoma soil nitrogen supply to plant nitrogen found in vegetation in high arctic Svalbard? I suggest using a more relevant plant nitrogen pool from Siberian or Alaskan graminoid tundra.

L254-256: Do you mean to say “short-term” instead of “long-term”? In the short-term, yes, plants will be disturbed when thaw slumps occur, but over longer time scales (decades+) then revegetation should occur in these permafrost subsided areas. These thaw slumps seem prime candidates for mixing of shallow and deeper permafrost layers, thereby potentially making at least some extent of the deeper nitrogen available for plants. Nevertheless, I agree with the authors that most deep permafrost nitrogen seems unlikely to become available to plants as it occurs too deep in the soil profile.

L257-259: I am not sure I fully understand what you mean, please rephrase. Microbial biomass is usually capable of taking up excess nitrogen when available. See various nitrogen fertilization studies in tundra sites.

L276: This seems contradictory to L179, which says that re-vegetated areas emitted the most N₂O.

L282-285: I am not convinced that all Yedoma deposits across the full depth range to 30-40 meters will be exposed to aerobic surface conditions if thawed. This might occur at erosion sites as

retrogressive thaw slumps where soil layers of many different depths and classes become mixed but otherwise the deep permafrost soils will continue to be overlain by many meters of soil, even if thawed. Deep soil layers will likely be anoxic, either because of water saturation at depth or because of oxygen decline with soil depth.

L289-304: So the conclusion is that landscape and hydrology changes upon thaw will determine N₂O production, but there is no quantification of the areal extent of landscapes, or soil depths or classes, that are most likely to contribute to N₂O production. Is it possible to add some estimate of this? This would reduce speculation and greatly add to the potential importance of Yedoma-wide N₂O production.

L291: Reference 53 relates to arctic river dynamics while the sentence is about microbial community effects on potential N₂O emissions from terrestrial permafrost landscapes?

L305:329: I thought this paragraph was difficult to get through. The conclusion is that nitrogen mobilization may either promote or deter organic matter decomposition?

L313:314: But these studies were on much different non-permafrost soils, right?

L326: I believe a reference would be appropriate here after “permafrost”.

L339: What do you mean by “likely also from Yedoma deposits”?

L346: Please specify the depth to help the reader.

L349: While I agree with the statement here, I find that this sentence contradicts with previous statement on L285 (see my comment above). In ice-rich permafrost, thawing will likely lead to inundated and anoxic deep soil layers due to water pooling at depth.

L362: Can you provide estimates of how much coastal and riverbank erosion areas cover of the total Yedoma domain? This seems relevant if these areas are the prime sites for deep permafrost thaw and mixing with overlying soil layers.

L368: This seems to contradict with L349, see comment above. Moreover, one aspect of N₂O production that I did not find discussed in the text is the likely consumption of N₂O by other microbes in overlying soil layers. I encourage the authors to discuss the possibility that much of any N₂O production at depth may be consumed in the active layer before it reaches the atmosphere. See, for example, Elberling et al. 2010 Nature Geoscience.

L372-374: Please rephrase. I do not think that I understand what you mean.

L382: This calculation seems rather speculative. Why estimate that 10% of the total full Yedoma nitrogen pool will be mobilized by 2100 when ¾ is stored at depths below 3 m? Can you provide some details on this estimate that make the assumption more realistic?

Methods: I would appreciate a bit more details on the depth data used for calculation of the deeper permafrost nitrogen content. In Table 2, the mean thickness is shown for each of the stratigraphic classes but how many distinct boreholes contributed to the deep permafrost class thickness and nitrogen data? This seems relevant as much of the estimated nitrogen content appears at depth. For example, the Yedoma class permafrost extends from near surface (<3 meter depth) to max depth (approximately 40 meter depth based on Figure 2) but it is unclear how many samples were obtained from deep layers.

Reviewer #2 (Remarks to the Author):

This paper updates the estimates of the amount of nitrogen in organic rich permafrost from the Yedoma domain, a region that contains about 25% of the world's total permafrost carbon. The new estimate suggests that N stocks in permafrost may have been underestimated by about 50%. The paper clearly lays out the 4 reasons why understanding the fate of N is important for long-term climate implications of permafrost thaw in the introduction (lines 62-83).

The authors analyzed over 2,000 soil samples from Alaska and Siberia for N content and mineral N concentrations. The samples were collected over a 20 year time frame and it is not clear from the methods how much of this data has been previously reported and how much of this is new. Are these all new analyses of stored samples, kept frozen?

While the paper does improve the global estimate of N in permafrost, I found the very long discussion fairly speculative and a review of previous work, including that by some of the authors. For example, the discussion on N₂O. At this point I think the role that thawing permafrost will play in N₂O emissions is very much an open question. Certainly, work by Marushchak and others has shown that at times there can be very significant N₂O emissions from areas where thawed permafrost has been revegetated. Others, such as Mu, have found a mix of results with some areas showing high N₂O emissions, and others very low emissions. Therefore, I'm not sure that the calculation on lines 379-392 is well justified or adds much beyond the discussion in the some of the original papers. The other sections are also largely reviews and do not rely on the data in this paper to improve our understanding of processes.

I think this work might be much better suited to a different journal as a larger review paper with more information about the soils themselves and some more direct comparisons to non-Yedoma permafrost. This would also allow for more discussion about possible regional variations for example and allow for larger supplementary tables with basic information about water content, etc. The review of previous work on the N in permafrost is well done and informative, it just does not reflect new insights based upon this work..

Other issues

TOC – were carbonates removed or not somehow was it determined they were not significant?

Table 1 and Figure 3 – The authors report a N density ranging from 0.9-2.2 10³kg m⁻³ - I think something is off in the units here. Table 1 gives a density of under 1 gm cm³, therefore a cubic meter of soil would weigh less than 106 gr or 103 kg. The N density given here would be greater than the total mass of the soil. Should this simply be Kg m⁻³? Although even then I get somewhat different numbers, using the bulk density for Yedoma deposits in Table 1 (0.87 g/cm⁻³) and the % N (0.36%) I get a number more like 3 Kg m⁻³ rather than 1.

Ammonium and nitrate - I am not quite clear how the authors obtained information from frozen soil units using a rhizon porewater sampler or porewater press. Presumably the soils were thawed? It would have been nice to have seen the data on water content to calculate a concentration for comparison to other systems and the references given on line 185 do not seem like the best ones, I was surprised Salmon et al. 2018 and other papers were not referenced instead. (Reference 41 on line 185 refers to a paper where samples were taken within the active layer not the permafrost layer and it not directly relevant. I was not able to get ref 52). While for an overall density the values do seem somewhat high it is still only a small % of the total N pool.

Manuscript "A globally relevant stock of soil nitrogen in the Yedoma permafrost domain" under review at Nature Communications

Point-by-point authors response (AR) to the reviewers comments (RC)

RC: Reviewer #1 (Remarks to the Author):

This study provides a detailed account of the nitrogen content in Yedoma permafrost-affected soils. Yedoma occurs in areas across Siberia and Alaska and this permafrost type is characterized by being ice-rich and containing organic matter that is considered relatively unprocessed and labile. It is estimated that the Yedoma permafrost domain stores about $\frac{1}{4}$ of the total permafrost carbon stock, while its nitrogen storage has been far less studied.

In this study, the authors impressively make use of more than 2000 samples, collected and analyzed from diverse landscapes and stratigraphic classes, to provide an updated estimate of total nitrogen and dissolved mineral nitrogen forms (NH_4 and NO_3) upscaled across the Yedoma domain to soil depths of up to 40 meters. I assume this max soil depth based on Figure 2 but I did not find it explicitly stated, while mean thickness for Yedoma class permafrost is given as 20 meters.

AR: Thank you for your review. For the bootstrapping we used the actual measured depth, and in Table 1 we give the mean depth of 20 metres. In the revised version we clarified this, e.g. as requested in the abstract and at fig.2. Here in short: Yes, Yedoma goes deep, sometimes 40-50 metres, but with bootstrapping technique the N pool is estimated down to ca. 20m for its deepest unit, the Yedoma deposits.

RC: The methods used and the resulting data generally appear robust, see specified comment below.

AR: Thank you very much. Please find our answers to your specified comments below.

RC: I believe this study will be a solid contribution to several research fields focusing on permafrost-affected landscapes. The amount of nitrogen stored in the Yedoma domain is large and I agree with the authors that this nitrogen stock is relevant in a global perspective and when considering various ecosystem-climate feedback mechanisms. However, the question remains how much of this permafrost nitrogen will become biologically available and mobilized in the future. The authors often convey a message of the Yedoma permafrost nitrogen stock to be vulnerable to thaw. Nevertheless, $\frac{3}{4}$ of the estimated nitrogen stock lies frozen below 3 meters and up to a max depth of several tens of meters.

Labeling near-surface and deep permafrost layers together as “vulnerable to thaw” seems somewhat unrealistic for the permafrost landscapes that are not erosion sites, e.g. not coastal or riverbed cliff areas. I believe that the manuscript will improve with some added text focusing on the extent of soil layers or depths, perhaps of specific landscape types, which are estimated to thaw over a certain period. This timeframe could be short- or long-term. Without some quantification of projected thaw trajectories and a timeframe, I find it difficult to assess the climate-feedback importance of the large nitrogen stock found at depth.

AR: Thank you for asking for clarification here. In general, we tried to include this point by Table 2 with its classification of the N pool according to age, depth and thaw legacy (L175).

For the erosional sites like riverside cliffs and coastal areas, deep mobilisation is possible and visually easy to recognize. Besides those sites, rapid thawing of deep permafrost also occurs underneath thermokarst lakes (e.g. Walter Anthony et al., 2018, <https://doi.org/10.1038/s41467-018-05738-9>) and at the margins of thermokarst basins, which are very abundant across the Yedoma domain (e.g. Nitze et al., 2018 <https://doi.org/10.1038/s41467-018-07663-3>). This is leading to mobilisation of deep N stocks also inland and widespread, which can then become available for plants and microbes. This process can mobilise tens of metres of sediments rapidly, i.e. over decades. Although these are local processes from a mechanistic point of view, this abrupt thaw is happening nearly everywhere in the Yedoma domain due to its high ice-content. Abrupt thaw of deep sediments causes both thaw lake expansion as well as rapid lake drainage, masking the change in thawed sediment volume. Along with changes in lake area occurring in both directions, the relatively small surface area change makes widespread assessment of Pan-Arctic changes difficult to detect using remote sensing data available over large regions. This has until now prevented accurate estimation of their total area across the Yedoma domain. Whether the thawed sediment volume translates to nitrogen mobility remains a crucial unknown.

One consideration (now explicitly included) is that ice wedges extend through the full depth of Yedoma. As soon as the ice melts, the soil slumps, water moves through soil deformities, and these processes mobilise the soil N. Via river networks, the N mobilised can be effectively spread over large areas, so the consequences of this added N are much broader than the degraded site itself.

In the revised manuscript, we added an extra paragraph discussing the vulnerability of deep N stocks, as you suggested (Line 79 – 90: “It is estimated that abrupt thaw processes could affect 1.6 million km² by 2100, impacting half of permafrost organic matter through collapsing ground, rapid erosion, and landslides(Turetsky et al. 2020). Especially Yedoma landscapes are one of the major reasons why so much permafrost carbon is stored in areas classified as “very high” risk of abrupt thaw and surface collapse in the future(Olefeldt et al. 2016). Because ice wedges extend through the full depth of Yedoma deposits (Fig.2), permafrost thaw triggers surface collapse as well as fluvial and coastal erosion (Turetsky et al. 2020; Farquharson et al. 2022; Irrgang et al. 2022). The melting of ice can also create macropores and microtopography that alter soil hydrology, potentially mobilising soil N from well below the surface(Farquharson et al. 2022). Thus, also deep mobilisation of OM is expected to intensify on the short- to mid-term term (coming decades(Kessler et al. 2012) to end this century(Turetsky et al. 2020)), in total up to twelve-fold faster than expected(Nitzbon et al. 2020). However, exact numbers are missing as in most models still the abrupt thaw processes are not implemented.”).

RC: Please see additional detailed commentary below.

Specific comments:

L16: Please add info on the depth limit of this nitrogen estimate.

AR: The depth data for Yedoma used for bootstrapping ranges between 5 to 41 m for the deepest unit Yedoma. For the bootstrapping technique we come (with our calculation) closest to the value of the mean depth, which is given now as suggested on the text (and table 1). In detail, we added this to the revised manuscript at various places (e.g. L40 (the former L16), L319) like the methods section, abstract and at a table caption). Moreover, we clarified Figure 2 to avoid misunderstandings.

RC: L19: I suggest reporting that $\frac{3}{4}$ of the Yedoma permafrost nitrogen is stored at depths below 3 m, or, conversely, report that $\frac{1}{4}$ is stored above 3 m. It would be strong to refer to estimates on which soil depth groups that will likely thaw within a certain timeframe. This could be short and/or long-term but it should be defined. Stating that permafrost nitrogen, which in this study is estimated down to several tens of meters, is vulnerable to thaw across the entire depth range seems rather i) unrealistic, and ii) vague.

AR: Thank you. We included the information on the portion below 3 m here (lines 43-46: “Here, we show that of this amount, $\frac{3}{4}$ is stored >3 metre depth, but if partially mobilised by thaw, this large nitrogen pool could have continental-scale consequences for soil and aquatic biogeochemistry and global-scale consequences for the permafrost feedback.”).

We do not completely agree with the “unrealistic” (see comment above) thaw vulnerability. As now included in the manuscript and answer to the reviewer comments below: please keep in mind that ice wedges extend through the full depth of Yedoma. As soon as the ice melts, the soil slumps, water moves through soil deformities, and these processes mobilise the soil N. Also, thermokarst lakes are one of the most efficient means of thawing permafrost deeply, and eroding at the basin margins also through the entire Yedoma deposits. Of course, this is not happening everywhere and at the same time. Actually, this was our motivation to separate the N stock by freezing state as well as depth in this paper (Table 2).

RC: L27: Please provide info on the depth limit of the carbon estimate here.

AR: We added the maximum depth (L53: “down to ~50 m) as described in the cited references.

RC: L32: Please specify what you mean by 4-6 % until 2300. It is not clear from the present text.

AR: We realised that this statement with the percentages was not ideal for the introduction. To make it easier to read we changed it to: “Permafrost thaw is exposing currently frozen, long dormant OM to microbial processes (up to mean loss of 341 Gt (~20%) of the now frozen OM

mineralised until 2300 under RCP8.5(McGuire et al. 2018)) and physical mobilisation (not fully quantified yet(Günther et al. 2013; Fuchs et al. 2020))”. (L56-59)

RC: L44: Please provide info on the depth limit of the carbon estimate here.

AR: The depth for Yedoma in the cited reference is bootstrapped. Thus, the depth is different for the units like thermokarst and Yedoma deposits. Nevertheless, we added “down to ~20m”, L 70) as suggested. Moreover, we added a level of detail for our depth data to the revised methods section.

RC: L54: Could you please add some details on the areal extent of deep soil (> 3 m) vulnerability to thaw in the coming decades (i.e. short-term outlook)?

AR: Schneider von Deimling et al. (2015, <https://doi.org/10.5194/bg-12-3469-2015>) analysed carbon fluxes from deep deposits separately for pristine Yedoma regions and refrozen thermokarst basins. The authors estimated that potentially 40 % (25%) of Yedoma (refrozen thermokarst) can be subject to new thermokarst lake formation. After thermokarst lake formation, sublake sediments can be thawed quickly (decadal timescale, Kessler et al. 2012, <https://doi.org/10.1029/2011jg001796>), but the moment of thermokarst lake formation is hard to predict as it depends on the depth of local excess ice presence in the ground. Following Turetsky et al. 2020 (<https://doi.org/10.1038/s41561-019-0526-0>) “Abrupt thaw will probably occur in more than 20% of the permafrost zone but could affect half of permafrost carbon through collapsing ground, rapid erosion and landslides”. Moreover, they state, “with warming and associated thaw, the total area of abrupt thaw increased to 1.6 million km² by 2100 and 2.5 million km² by 2300.”

We emphasise these findings about the potential for release of N below 3 m in the revised manuscript (L79-90): “It is estimated that abrupt thaw processes could affect 1.6 million km² by 2100, impacting half of permafrost organic matter through collapsing ground, rapid erosion, and landslides(Turetsky et al. 2020). Especially Yedoma landscapes are one of the major reasons why so much permafrost carbon is stored in areas classified as “very high” risk of abrupt thaw and surface collapse in the future(Olefeldt et al. 2016). Because ice wedges extend through the full depth of Yedoma deposits (Fig.2), permafrost thaw triggers surface collapse as well as fluvial and coastal erosion (Turetsky et al. 2020; Farquharson et al. 2022; Irrgang et al. 2022). The melting of ice can also create macropores and microtopography that alter soil hydrology, potentially mobilising soil N from well below the surface(Farquharson et al. 2022). Thus, also deep mobilisation of OM is expected to intensify on the short- to mid-term term (coming decades(Kessler et al. 2012) to end this century(Turetsky et al. 2020)), in total up to twelve-fold faster than expected(Nitzbon et al. 2020). However, exact numbers are missing as in most models still the abrupt thaw processes are not implemented”

RC: L74: These studies (ref.: 39 and 40) do not focus on permafrost-affected soils.

AR: Thanks, we are aware of that. This sentence was focused on OM decomposition, not necessarily in permafrost regions, and we cited these studies to include the spectrum of pathways and processes. To avoid misunderstanding we deleted these 2 citations and changed the sentence to “While some studies found that greater N availability can accelerate OM decomposition(Mack et al. 2004; Chen et al. 2018), at least for the first years after thaw(Salmon et al. 2018), others show the opposite effect thereby either amplifying or

reducing the climate feedback depending on the net interaction(Shaver et al. 2000; Beermann et al. 2015; Koven et al. 2015; McGuire et al. 2018; Wologo et al. 2021).” (L108-111)

RC: L85/research goal ii: I agree that assessing quantity and vulnerability of permafrost nitrogen is timely and important. However, I am not convinced that the present study fully achieves the “vulnerability assessment” aspect. Without some quantification of the extent of both shallow and deep permafrost (> 3 m) nitrogen that is estimated to thaw by a specific time, some of the statements about substantial climate feedbacks seem speculative.

AR: Thank you for pointing out the importance of the research question related to the expected feedback effect between the soil N stocks and the climate. We understand vulnerability as a summed effect of at least two different characteristics: 1) how vulnerable the N contained in different landscape units is for physical mobilisation with permafrost thaw?, and 2) how bioavailable is this N for plants and microorganisms?

Maybe you see the vulnerability as probability of permafrost thaw and physical liberation of N with this thaw. Which is of course right, but besides this, we try to clarify how the **quality** of Yedoma OM and N is making it vulnerable for mobilisation. We mean here that Yedoma N is likely highly bioavailable, as indicated by high N content, low C/N ratio and low DON/mineral N ratio. These are all original results from this study and not part of the review part of the former discussion. To make this point clearer we shortened the discussion significantly to focus on our finding. Moreover we added the abovementioned vulnerability statement to the discussion (L294-297: “These large N stocks also appear to be highly bioavailable as we found that the quality of Yedoma domains’ OM is making it vulnerable/favourable for mobilisation, as its N is potentially highly bioavailable, as indicated our results of high N content, low C/N ratio and low DON/mineral N ratio.”)

RC: Research goal ii states that the goal of this study is to “... improve our understanding of the susceptibility of these N reservoirs to thaw and climate change”. While the study successfully quantifies Yedoma nitrogen stocks (research question i), the remaining text generally reviews literature on the lability of Yedoma organic matter and possible ecosystem effects of nitrogen mobilization. This seems a somewhat different focus than what is actually stated in research goal ii. I suggest changing “improve” to “review” in research goal ii.

AR: In general, this paper was oriented towards starting a discussion and presenting a large dataset for further research. To follow your feedback, we specified the goal to “(ii) to discuss these results with previously published data to enhance the understanding of potential climate feedbacks”. (L123-124). This in mind we also shortened the review part of the discussion substantially.

RC: Figure 1: When printed (but also to some extent digitally), it is not always easy to distinguish the green colour for “tree line” from the blue Yedoma distribution range. Consider modifying colour scheme.

AR: Changed accordingly (treeline now green, and tested with <https://www.color-blindness.com/cobli-color-blindness-simulator/> for different colour vision deficiencies).

RC: Figure 2: This is a nice overview conceptual figure. I recommend inverting the Y-axis so that 0 m is at the ground surface. Please provide some explanation for the gray thermokarst

colour class, and explanation for the gray dashed line that shows the extent of the Taberite class. How are thermokarst depressions used in the depth calculations, do they begin at several meters of depth or do you count the surface of thermokarsts as 0 meter, similar to Yedoma upland? This info seems relevant for the depth-specific nitrogen estimates. Note that Taberite is supposedly purple according to the text but appears blue on my screen and in print. Using the colour scheme here for the subsequent figures and tables works well. However, I encourage the authors to use a colourblind-friendly colour scheme. I do not think that the “active layer” class needs to be red if thermokarst is green.

AR: Thank you for all these recommendations. We adapted the colour codes in all figures focussing on red/green deficiencies. Moreover, we changed the altitude to depth as suggested, and the formerly grey sediment is now part of the active layer class. We deleted the grey dashed line. For the depth, we added an explanatory sentence to the caption, as the depth there is just for a scale (L150-152, “The depth is approximate and meant to communicate the general depth of different landscape units. For the depth-specific nitrogen estimates, absolute depth measurements are used.”)

RC: L177: Are there any model estimates in recent literature of how much of this deep permafrost (>3 m depth) that will be exposed to thaw over a specified timeframe? That seems relevant here for the importance of the deep soil nitrogen pools.

AC: This would be great, but because of the complications of simulating abrupt thaw and surface reorganisation, there are not adequate estimates of the volume of permafrost thawed at a circumpolar scale. For example, Turetsky et al (2020) synthesised that after considering abrupt thaw stabilisation, lake drainage and soil carbon uptake by vegetation regrowth, models considering only gradual permafrost thaw are substantially underestimating carbon emissions from thawing permafrost. Similarly, Cheng et al. (2022, <https://doi.org/10.1038/s41467-022-29011-2>) estimate areal rates in alpine areas, but this does not resolve the question of how deeply the thaw extends. We have expanded the text to include discussion of these uncertainties and the likelihood that extent and depth of thaw are largely underestimated. We did this e.g. in the introduction: “Thus, also deep mobilisation of OM is expected to intensify on the short- to mid-term term (coming decades(Kessler et al. 2012) to end this century(Turetsky et al. 2020)), in total up to twelve-fold faster than expected(Nitzbon et al. 2020). However, exact numbers are missing as in most models still the abrupt thaw processes are not implemented.” (L87-90), or in the discussion: “Looking at the influence of thaw processes in the Yedoma domain on N stocks we found that thaw processes in the Yedoma domain can activate deep soil deposits through mechanisms such as thermo-erosion, coastal collapse, fluvial erosion, and thermokarst lake formation(Schneider von Deimling et al. 2015; Fuchs et al. 2020; Runge et al. 2022). Besides being confident that models considering only gradual permafrost thaw are substantially underestimating carbon emissions from thawing permafrost(Schneider von Deimling et al. 2015; Nitzbon et al. 2020; Turetsky et al. 2020), no quantitative estimate on future thaw depth and coverage are available.” (L235-240).

RC: L180: But that depends on how thaw occurs, right? With thaw collapse as in retrogressive thaw slumps then shallow and deeper permafrost layers may be mixed and some of the deep Yedoma nitrogen could contribute to relatively high N₂O fluxes as in the study by Marushchak

et al. However, in other landscapes, where thaw slumping does not occur as drastically then deeper permafrost layers will likely stay buried and anoxic, which may not promote N₂O production. See also comment for L282.

AR: As mentioned before we shortened the discussion significantly to focus on our key results. This part of the discussion is not included any more.

Nevertheless, we want to answer your concern: You are right that N₂O emissions will likely not occur everywhere where Yedoma permafrost thaws (Voigt et al., 2020 <https://doi.org/10.1038/s43017-020-0063-9>). In thermokarst lakes, for example, the anoxic conditions will favour strictly anaerobic metabolism, including methanogenesis. However, it is important to note that ice wedges extend through the full depth of Yedoma. As soon as the ice melts, the soil slumps, water moves through soil deformities, and these processes mobilise the soil N at depth. With high nutrient content (e.g. nitrate), N₂O may be released even under anoxic conditions. These deep disturbance processes are local on the process scale, but they are widespread, especially in the Yedoma domain. In addition, we must also consider the erosion processes in the margins of the countless thermokarst lakes and basins.

RC: L197: Please check your values, plant nitrogen per m² should not be this high. In addition, why compare Yedoma soil nitrogen supply to plant nitrogen found in vegetation in high arctic Svalbard? I suggest using a more relevant plant nitrogen pool from Siberian or Alaskan graminoid tundra.

AR: We agree that the previously cited value from Svalbard was not the best for this paper. We found more representative range of plant uptake from Wild et al. (2018) who summarise the relevant published literature: “We thus arrived at a maximum daily plant N uptake of 83 mg N m⁻² d⁻¹ during the growing season (total range 1.7–83mg N m⁻² d⁻¹).” This makes maximum plant uptake of 0.17–8.3 g N m⁻² yr⁻¹ assuming 100 growing season days. Furthermore, they cited Chapin et al. (1988) 0.8 and 1.5 gNm⁻² yr⁻¹ in Alaskan tussock tundra, which is lower but considered aboveground biomass only and did not take into account the N uptake to the belowground parts. Despite further literature research, we could not find any values for the Siberian High-Arctic.

Anyway, due to the shortening of the discussion this paragraph is not part of the revised manuscript any more.

RC: L254-256: Do you mean to say “short-term” instead of “long-term”? In the short-term, yes, plants will be disturbed when thaw slumps occur, but over longer time scales (decades+) then revegetation should occur in these permafrost subsided areas. These thaw slumps seem prime candidates for mixing of shallow and deeper permafrost layers, thereby potentially making at least some extent of the deeper nitrogen available for plants. Nevertheless, I agree with the authors that most deep permafrost nitrogen seems unlikely to become available to plants as it occurs too deep in the soil profile.

AR: We thank you for pointing this out and agree that this text part was wrong, as we meant short-term as you suggested. Due to shortening the discussions this part is not in the revised manuscript any more.

RC: L257-259: I am not sure I fully understand what you mean, please rephrase. Microbial biomass is usually capable of taking up excess nitrogen when available. See various nitrogen fertilization studies in tundra sites.

AR: That is correct. We state that the released N is susceptible to microbial processes including nitrification, denitrification (N_2O and N_2 production), and a whole suite of other processes, including immobilisation into the microbial biomass. Shortly after thawing, net mineralisation rates were found to be low; resulting in immobilisation of N in the microbial biomass, but N mineralisation rates increased a few years after thawing resulting in excess mineral N and N_2O production due to the high content and bioavailability in thawed Yedoma permafrost (Marushchak et al. 2021).

Due to shortening the discussion, this part is not in the revised manuscript any more.

RC: L276: This seems contradictory to L179, which says that re-vegetated areas emitted the most N_2O .

AR: We agree that clarification would have been needed for both statements. Due to shortening the discussion, this part is not in the revised manuscript any more.

Anyway, we want to fully answer the reviewers comment: When considering permafrost-affected soils in general high N_2O emissions usually occur from areas where vascular plants are absent or disturbed (due to lower plant N uptake compared to vegetated areas). In eroding Yedoma, however, the highest N_2O emissions were found after pioneering plants had started to re-establish after thaw, potentially due to the time needed to drain the soil and accumulate oxidised inorganic N (e.g., NO_3 and NO_2). Higher N_2O emissions were related to slope stabilisation, soil drying, C-inputs due to plants, and structural changes in the microbial N-cycling community, which co-occurred with re-vegetation (very wet conditions immediately after thaw limit nitrification and may cause reduction of atmospheric N_2O to N_2).

RC: L282-285: I am not convinced that all Yedoma deposits across the full depth range to 30-40 meters will be exposed to aerobic surface conditions if thawed. This might occur at erosion sites as retrogressive thaw slumps where soil layers of many different depths and classes become mixed but otherwise the deep permafrost soils will continue to be overlain by many meters of soil, even if thawed. Deep soil layers will likely be anoxic, either because of water saturation at depth or because of oxygen decline with soil depth.

AR: Yes, right, and we do not want to say that here. To make this clearer we added the word “partially” (L272) to tone down our statement, as well as a sentence including this depth limitation (“As mentioned before not all Yedoma deposits all across the full depth range will be exposed if thawed” (273-274). For the other aspects, please see our comments on the deep thaw vulnerability because of excess ice melting above.

In principle, N_2O can be released under anaerobic conditions also, if nutrients (particularly nitrate) are high. This has been shown e.g. in boreal lakes, which could be sources of N_2O (mostly the nutrient rich lakes). Please see this reference here: <https://onlinelibrary.wiley.com/doi/full/10.1111/gcb.14928>. However, of course it is likely that all these processes will be less prominent, and/or that N_2O is reduced to N_2 under

anaerobic conditions, that diffusion is too slow. To keep the discussion shortened we did not include this additional discussion in the revised manuscript.

RC: L289-304: So the conclusion is that landscape and hydrology changes upon thaw will determine N₂O production, but there is no quantification of the areal extent of landscapes, or soil depths or classes, that are most likely to contribute to N₂O production. Is it possible to add some estimate of this? This would reduce speculation and greatly add to the potential importance of Yedoma-wide N₂O production.

AR: See our replies to the RCs above. Unfortunately, there is no estimate on the areal extent of thaw. Due to shortening the discussion, this part is not in the revised manuscript any more.

RC: L291: Reference 53 relates to arctic river dynamics while the sentence is about microbial community effects on potential N₂O emissions from terrestrial permafrost landscapes?

AR: Thanks for detecting this mistake. The right reference would have been Marushchak et al. 2021, but due to the shortening, this part is not in the revised manuscript any more.

RC: L305:329: I thought this paragraph was difficult to get through. The conclusion is that nitrogen mobilization may either promote or deter organic matter decomposition?

AR: Thank you for this input. We have taken this paragraph out of the revised discussion.

RC: L313:314: But these studies were on much different non-permafrost soils, right?

AR: Yes, they are on non-permafrost soils. As it was a statement on the processes, these were, to our mind, helpful references. In the revised version with the streamlined discussion this references are not included anymore.

RC: L326: I believe a reference would be appropriate here after “permafrost”.

AR: We would have added Ewing et al. 2015 (<https://doi.org/10.1002/2015gl066296>), right, but this part was delete while shortening the discussion.

RC: L339: What do you mean by “likely also from Yedoma deposits”?

AR: We meant the Yedoma in the catchments. This part was deleted while shortening the discussion.

RC: L346: Please specify the depth to help the reader.

AR: We meant deeper than 3 m, but deleted also this part for streamlining the discussion.

RC: L349: While I agree with the statement here, I find that this sentence contradicts with previous statement on L285 (see my comment above). In ice-rich permafrost, thawing will likely lead to inundated and anoxic deep soil layers due to water pooling at depth.

AR: We deleted this part in the discussion. To answer you comment this phrasing would have been clearer: “The deeper strata are likely to remain anoxic following thaw if it is not associated with changes in hydrology; however, as discussed above, thaw and ground subsidence can often lead to surface exposure and improved drainage of deep sediment layers (on thaw slumps and drained thermokarst lake basins).”

RC: L362: Can you provide estimates of how much coastal and riverbank erosion areas cover of the total Yedoma domain? This seems relevant if these areas are the prime sites for deep permafrost thaw and mixing with overlying soil layers.

AR: There is no valid simple way to estimate the coverage of these areas. This is an entire project on its own. For coasts, taking an updated ACD (Arctic Coastal Dynamics) database could be a future way to start, but there is nothing for river erosion bluffs. Like above, this commented part is not included in the revised manuscript any more.

RC: L368: This seems to contradict with L349, see comment above.

AR: Thank you. We skipped this statement, considering it is too speculative.

RC: Moreover, one aspect of N₂O production that I did not find discussed in the text is the likely consumption of N₂O by other microbes in overlying soil layers. I encourage the authors to discuss the possibility that much of any N₂O production at depth may be consumed in the active layer before it reaches the atmosphere. See, for example, Elberling et al. 2010 Nature Geoscience.

AR: Thank you for bringing up this. Due to focussing more on our own data, we skipped this topic in the revised discussion.

Nevertheless, you are right; a full N feedback is needed. Of course, we are aware of the citation, which was already cited elsewhere in the manuscript. In this context we find it is very important to emphasise that although it matters a lot for the direct climate forcing if N is released as the strong GHG N₂O or atmospherically inert N₂, any losses of nitrogenous gases mean that this N will not be available for plant growth. This may cause limitations for plant growth and CO₂ fixation from the atmosphere.

RC: L372-374: Please rephrase. I do not think that I understand what you mean.

AR: We deleted this text.

RC: L382: This calculation seems rather speculative. Why estimate that 10% of the total full Yedoma nitrogen pool will be mobilized by 2100 when $\frac{3}{4}$ is stored at depths below 3 m? Can you provide some details on this estimate that make the assumption more realistic?

AR: We agree, of course this is speculative. Following your and the other reviewers suggestions we decided to shift back closer to the findings of our study. This calculation is not part of the revised version any more.

Nevertheless, we also want to answer your comment properly: This was a back-of-the-envelope calculation, yes, which was added in order to help readers understand the rough magnitude of the potential ecological importance of this added N from permafrost thaw. The 10% assumption is referenced for permafrost in general (Schuur et al.), which we assumed to be the same for Yedoma domain. As mentioned in the initial text, this rough estimate follows an approach by Ramm et al. (2020, <https://doi.org/10.1007/s00376-020-0027-5>), updated with the latest available numbers from our study. As mentioned earlier in our response, abrupt processes can reach great depths, and with erosion, thermokarst lake development and

draining thermo-erosional valleys plus gradual active layer deepening and associated surface subsidence this number is, in our opinion, realistic.

RC: Methods: I would appreciate a bit more details on the depth data used for calculation of the deeper permafrost nitrogen content. In Table 2, the mean thickness is shown for each of the stratigraphic classes but how many distinct boreholes contributed to the deep permafrost class thickness and nitrogen data? This seems relevant as much of the estimated nitrogen content appears at depth. For example, the Yedoma class permafrost extends from near surface (<3 meter depth) to max depth (approximately 40 meter depth based on Figure 2) but it is unclear how many samples were obtained from deep layers.

AR: We added the depth information to the methods (number of sites, range and mean/median: “The depth with a complete outcropped sediment unit (full range, top to bottom) was taken from 21 sites for Yedoma deposits (range 5 – 41 m, mean 20 m, median 19 m) and 9 sites for thermokarst deposits (range 2 – 13 m, mean 6 m, median 5 m) (L317-320).

The majority of the Yedoma data is below 3 m. Due to an inhomogeneity of depth data (in many cases it is **height** from bluffs, starting with 0 m at the river bank/coast without a possibility to set the exact base depth due to inaccessibility) it is just an approximation which samples are below 3 m for Yedoma. The dataset in total is submitted to PANGAEA and will be fully available.

RC: Reviewer #2 (Remarks to the Author):

This paper updates the estimates of the amount of nitrogen in organic rich permafrost from the Yedoma domain, a region that contains about 25% of the world's total permafrost carbon. The new estimate suggests that N stocks in permafrost may have been underestimated by about 50%. The paper clearly lays out the 4 reasons why understanding the fate of N is important for long-term climate implications of permafrost thaw in the introduction (lines 62-83).

The authors analyzed over 2,000 soil samples from Alaska and Siberia for N content and mineral N concentrations. The samples were collected over a 20 year time frame and it is not clear from the methods how much of this data has been previously reported and how much of this is new. Are these all new analyses of stored samples, kept frozen?

AR: The samples were measured after being freeze-dried (added to the methods section, L333) after the expeditions. We included newly measured samples and samples from unpublished databases from our labs. The samples were neither published with an N-focus on nitrogen, nor used for stock estimation. In this context, this data is new.

RC: While the paper does improve the global estimate of N in permafrost, I found the very long discussion fairly speculative and a review of previous work, including that by some of the authors. For example, the discussion on N₂O. At this point I think the role that thawing permafrost will play in N₂O emissions is very much an open question.

AR: Thank you for mentioning the improvement of our findings. We followed your suggestion and streamlined the discussion, especially on the potential fate of nitrogen.

We focus now on our own data, as the revised discussion highlights our new nitrogen stock data and combines them with our ammonium and nitrate data. Further, we provide context with assumptions and previously published data. We clarified the new findings, especially about how the quality of Yedoma organic matter and nitrogen makes it vulnerable for mobilisation. Since these are all original results from this study, binding the discussion closer to our own data was necessary to address your major revision.

RC: Certainly, work by Marushchak and others has shown that at times there can be very significant N₂O emissions from areas where thawed permafrost has been revegetated. Others, such as Mu, have found a mix of results with some areas showing high N₂O emissions, and others very low emissions. Therefore, I'm not sure that the calculation on lines 379-392 is well justified or adds much beyond the discussion in the some of the original papers.

AR: We agree. Of course, this was speculative; nevertheless, we clearly highlighted it as a rough estimate in the initial manuscript. Following your and the other reviewers suggestions we decided to shift back closer to the findings of our study. This calculation is not part of the revised version any more.

RC: The other sections are also largely reviews and do not rely on the data in this paper to improve our understanding of processes.

AR: We hope that we included your suggestion after substantially shortening our discussion. This revised discussion uses our new stock data, combines it with both NH₄⁺ and NO₃⁻, and put

this into context with assumptions and rates published. Moreover, we clarified the new findings, especially how the quality of Yedoma OM and N is making it vulnerable for mobilisation. For instance, the Yedoma N is likely highly bioavailable, as indicated by high N content, low C/N ratio and low DON/mineral N ratio.

RC: I think this work might be much better suited to a different journal as a larger review paper with more information about the soils themselves and some more direct comparisons to non-Yedoma permafrost. This would also allow for more discussion about possible regional variations for example and allow for larger supplementary tables with basic information about water content, etc. The review of previous work on the N in permafrost is well done and informative, it just does not reflect new insights based upon this work.

AR: Thank you for stating that the initial discussion is well done and informative, but too much a synthesis in its entire length. As mentioned above we rewrote the discussion.

We are convinced that our revised manuscript with our found nitrogen stock brings new insights of broad interest for the reader of Nature Communications. In detail, we showed that the Yedoma domain N stock increases the previous estimate of N stocks for the entire permafrost zone by nearly 50%. We found that approximately 90% of the N frozen, and therefore currently immobile. Even with only a partial mobilisation, fluxes from this large N pool could have continental-scale consequences for soil and aquatic biogeochemistry and global-scale consequences.

Moreover, we think that this paper fits well into this journal. It aligns well with the recently launched Nature collection “Permafrost in a warming world” (<https://www.nature.com/collections/ababhhdce/>). In this Nature collection there is one paper on nitrous oxide emissions from permafrost-affected soils (<https://www.nature.com/articles/s43017-020-0063-9>). Recently, in Nature Communications, there was a study presenting N₂O field observations from two Siberian Yedoma sites suggesting that this type of permafrost is of crucial importance for understanding the future permafrost nitrogen cycle, including the potent greenhouse gas N₂O (<https://www.nature.com/articles/s41467-021-27386-2>).

Expanding from these site-level studies, our upscaling study now provides a first glimpse into the total N pool affected by such processes.

RC: Other issues

TOC – were carbonates removed or not somehow was it determined they were not significant?

AR: Yes, carbonates were removed. We added this to the methods (L331)

RC: Table 1 and Figure 3 – The authors report a N density ranging from 0.9-2.2 103kg m⁻³ - I think something is off in the units here. Table 1 gives a density of under 1 gm cm³, therefore a cubic meter of soil would weigh less than 106 gr or 103 kg. The N density given here would be greater than the total mass of the soil. Should this simply be Kg m⁻³? Although even then I get somewhat different numbers, using the bulk density for Yedoma deposits in Table 1 (0.87 g/cm⁻³) and the % N (0.36%) I get a number more like 3 Kg m⁻³ rather than 1.

AR: You are completely right. The numbers were correct, but we falsely wrote 10^3 kg/m³ instead of kg/m³ for the N density. We corrected it, thanks.

RC: Ammonium and nitrate - I am not quite clear how the authors obtained information from frozen soil units using a rhizon porewater sampler or porewater press. Presumably the soils were thawed?

AR: Yes, they were thawed. We added this information to the methods.

RC: It would have been nice to have seen the data on water content to calculate a concentration for comparison to other systems and the references given on line 185 do not seem like the best ones, I was surprised Salmon et al. 2018 and other papers were not referenced instead. (Reference 41 on line 185 refers to a paper where samples were taken within the active layer not the permafrost layer and it not directly relevant. I was not able to get ref 52). While for an overall density the values do seem somewhat high it is still only a small % of the total N pool.

AR: Salmon et al 2018 was cited as the initial citation 38 already (Salmon, V. G. et al., <https://doi.org/10.1029/2018JG004518> (2018)). As well as with reference (initial submitted version) 47 (Salmon, V. G. et al., doi:10.1111/gcb.13204 (2016)) we added both also there as we agree that these references help here as well. With Beermann et al. 2017 paper (<https://doi.org/10.1002/ppp.1958>), we are happy to share a free copy (not typeset author version), if you have no access.

Following your advice we changed the sentence and references to: "High mineral N content has also been observed in Yedoma domain permafrost previously (of NH₄⁺ in particular, but also of NO₃⁻) (Beermann et al. 2015; Salmon et al. 2016; Beermann et al. 2017; Salmon et al. 2018), which could be due to a combination of SOM decomposition prior to permafrost aggradation, during the frozen period, or after permafrost degradation." (L227-230)

Concerning the water data, we included a sentence to the method section (L355-358).

Moreover, we do agree with the reviewer that the mineral N content of the Yedoma sediments is a small % of the total N pool. However, this is the case in any pristine northern soil, particularly for permafrost-affected soils, which often do not show measurable amounts of mineral N. Measurable, and often high mineral N content in Yedoma permafrost can be interpreted as a legacy of high reactive N availability in these sediments before permafrost aggradation and/or accumulation of inorganic N while frozen. Thus, this could be also as indicators of high potential for N mineralisation following thaw. With the mineral N figures, we wanted to highlight the differences with the various landscape units studied here and allow comparison with literature values.

REVIEWER COMMENTS

Reviewer #1 (Remarks to the Author):

This is a resubmission of a manuscript that I previously reviewed. The revised manuscript benefits from omitting much of the speculative text previously found in the Discussion section. I also welcome the addition of text focusing on potential vulnerability of deep permafrost N and unknowns about future thaw depths. While I find these additions helpful, the manuscript would still benefit greatly from some form of quantification of the future extent of thawed permafrost N considering that $\frac{3}{4}$ of the Yedoma N pool lies below 3 m. I understand that most models do not incorporate abrupt thaw or excess ice processes and that the different stratigraphic units are not observable fully using remote sensing products. Nevertheless, some model studies do simulate permafrost thaw depths in discrete permafrost landscapes, such as for ice-rich lake basins and Yedoma and Holocene deposits, and including excess ice and abrupt thaw processes. See for example Nitzbon et al. 2020 Nature Communications, which provide mean thaw depth outputs by 2100 for distinct ice-rich landscapes in NE Siberia under different climate change scenarios and including and excluding waterlogging. Adding even just a relatively crude “best estimate” calculation based on, for example, such thaw depth projections, would go a long way in convincing the reader about the size of the Yedoma N pool that will be thawed in the future.

Other commentary.

Line 53 – The wording here makes it sound as if the 1500 Gt OC was quantified down to 50 m depth across the entire permafrost region.

Line 124 – Is manuscript research goal 2 “to discuss these results with previously published data to enhance the understanding of potential climate feedbacks” different from what is normally done in a Discussion section, i.e. discuss results and relate findings to previous studies?

Line 228 – This seems a feature also found in non-Yedoma permafrost landscapes, see for example Fouche et al. 2020 Nature Communications.

Reviewer #2 (Remarks to the Author):

The authors have made substantial revisions to the manuscript, greatly improving and shortening the discussion. Importantly, they have decreased the parts of the discussion which were very speculative. They have also made the figures easier to read. They have provided comprehensive point by point responses to all of the questions raised by both myself and the other reviewer. They have clarified that the work brings together data on soil N content from more than 2000 samples that was not previously available. While I could see this paper in a different journal with more cross comparisons to other systems I do agree that it is a strong contribution on a very important topic.

Manuscript "A globally relevant stock of soil nitrogen in the Yedoma permafrost domain" under review at Nature Communications

Point-by-point authors response (AR) to the reviewers comments (RC)

RC: Reviewer #1 (Remarks to the Author):

This is a resubmission of a manuscript that I previously reviewed. The revised manuscript benefits from omitting much of the speculative text previously found in the Discussion section. I also welcome the addition of text focusing on potential vulnerability of deep permafrost N and unknowns about future thaw depths.

AR: Thank you very much for re-reviewing our revised version and for the positive feedback.

RC: While I find these additions helpful, the manuscript would still benefit greatly from some form of quantification of the future extent of thawed permafrost N considering that $\frac{3}{4}$ of the Yedoma N pool lies below 3 m. I understand that most models do not incorporate abrupt thaw or excess ice processes and that the different stratigraphic units are not observable fully using remote sensing products. Nevertheless, some model studies do simulate permafrost thaw depths in discrete permafrost landscapes, such as for ice-rich lake basins and Yedoma and Holocene deposits, and including excess ice and abrupt thaw processes. See for example Nitzbon et al. 2020 Nature Communications, which provide mean thaw depth outputs by 2100 for distinct ice-rich landscapes in NE Siberia under different climate change scenarios and including and excluding waterlogging. Adding even just a relatively crude “best estimate” calculation based on, for example, such thaw depth projections, would go a long way in convincing the reader about the size of the Yedoma N pool that will be thawed in the future.

AR: We appreciate this suggestion for including a best estimate on the N which will thaw in future. As you suggested to apply the approach by Nitzbon et al. (2020) we now include Jan Nitzbon in the author team. We added a paragraph to the discussion summarizing the results:

Lines 262-265: “A first-order estimate following the approach of Nitzbon et al.²⁷ suggests that by 2100 an additional 0.2–0.8 Gt N could be affected by thaw in the Yedoma region under an ambitious mitigation scenario (RCP2.6), and 4.3–16.3 Gt N – about 40% of the total N stock estimate – under a high emission scenario (RCP8.5).”

as well as a description of the method in the method section:

*Lines 396-406: “**Projection of thaw-affected nitrogen stocks***

We followed Nitzbon et al.²⁷ to estimate the portion of the N stocks that could become subject to thawed conditions within the course of the current century. For this, we took the maximum thaw depths simulated by Nitzbon et al.²⁷ for two different geomorphological units (Thermokarst lake basins and Yedoma uplands) using climatic forcing data for the central Lena River delta under three different emission scenarios (Representative Concentration Pathway (RCP) 2.6, RCP 4.5, and RCP 8.5) and under contrasting hydrological conditions (water-logged and well-drained). The simulations took into account rapid thaw processes due to the presence of excess and wedge ice. We multiplied the simulated thaw depths with the areal coverage of

the respective geomorphological units and the N densities of the corresponding stratigraphic classes that would be affected by thaw (Table 1)."

According to Nitzbon et al we included a figure on the projection of thaw-affected N stocks (now figure 5, Line 269-273)

Fig. 5 Projection of thaw-affected N stocks. The model estimate for thaw-affected C stocks by Nitzbon et al.²⁷ was adjusted to our N stocks by scaling simulated thaw depths under consideration of excess ice melt with the total areal extent of three geomorphological units composed of five stratigraphical classes (see Table 1, Fig. 2). For each warming scenario the indicated ranges correspond to 11-year running means of the annual maximum of thawed N under contrasting hydrological conditions.

Moreover we included an additional important reference (released 19 July 2022) to underline the relevance of the newly thawed N pool:

Lines 275-283: "If available to plants, this newly thawed N of up to ~16 Gt (high emission scenario) could have a fertilization effect and increase plant growth. However, this fertilization effect was recently found to be much weaker than expected in permafrost ecosystem in a modelling study (Lacroix et al. 2022). This was likely caused by a significant mismatch between the timing of peak plant growth (early to mid-summer) and peak thaw depth (late summer to fall) that resulted in incomplete plant use of N deeper in the profile, near the permafrost table. The increased N availability enhanced the N loss pathways, leading to increased N₂O emissions the applied model (Lacroix et al. 2022). With our data, this could mean that the deeper the thaw, the greater the temporal mismatch and increasing potential for N₂O release."

reference:

Lacroix et al. 2022: <https://doi.org/10.1111/gcb.16345>

RC: Other commentary.

Line 53 – The wording here makes it sound as if the 1500 Gt OC was quantified down to 50 m depth across the entire permafrost region.

AR: Yes, we changed that to make clear that this inclusion of deeper deposits was just done for some selected deposits (Yedoma and deltaic deposits, for example), not in general. As we

define Yedoma later, we avoided to bring the term Yedoma up here already. In detail, we added “for some deep deposits” (Line 53).

RC: Line 124 – Is manuscript research goal 2 “to discuss these results with previously published data to enhance the understanding of potential climate feedbacks” different from what is normally done in a Discussion section, i.e. discuss results and relate findings to previous studies?

AR: The intention was to set the focus on enhancing the *understanding of potential climate feedbacks*. Following this suggestion revised this to “Our specific goals are to estimate the N pools across different stratigraphic units of Yedoma domain permafrost soils and deposits (Fig.1) to enhance the understanding of potential climate feedbacks.” (Lines 124-126)

RC: Line 228 – This seems a feature also found in non-Yedoma permafrost landscapes, see for example Fouche et al. 2020 Nature Communications.

AR: Thank you. We added this statement and citation to the manuscript (Line 226).

RC: Reviewer #2 (Remarks to the Author):

The authors have made substantial revisions to the manuscript, greatly improving and shortening the discussion. Importantly, they have decreased the parts of the discussion which were very speculative. They have also made the figures easier to read. They have provided comprehensive point by point responses to all of the questions raised by both myself and the other reviewer. They have clarified that the work brings together data on soil N content from more than 2000 samples that was not previously available. While I could see this paper in a different journal with more cross comparisons to other systems I do agree that it is a strong contribution on a very important topic?

AR: Thank you for your reviews and feedback. We are very pleased that you agree that our work is a strong contribution on a very important topic.